# Foxc1 dependent mesenchymal signalling drives embryonic cerebellar growth

**Parthiv Haldipur[1], Gwendolyn S Gillies[1], Olivia K Janson[1], Victor V Chizhikov[2], Divakar S Mithal[3], Richard J Miller[3], Kathleen J Millen[1,4]***

[1]Center for Integrative Brain Research, Seattle Children's Research Institute, Seattle, United States; [2]Department of Anatomy and Neurobiology, University of Tennessee Health Sciences Center, Memphis, United States; [3]Department of Molecular Pharmacology and Biological Chemistry, Northwestern University, Chicago, United States; [4]Department of Pediatrics, Genetics Division, University of Washington, Seattle, United States

**Abstract** Loss of *Foxc1* is associated with Dandy-Walker malformation, the most common human cerebellar malformation characterized by cerebellar hypoplasia and an enlarged posterior fossa and fourth ventricle. Although expressed in the mouse posterior fossa mesenchyme, loss of *Foxc1* non-autonomously induces a rapid and devastating decrease in embryonic cerebellar ventricular zone radial glial proliferation and concurrent increase in cerebellar neuronal differentiation. Subsequent migration of cerebellar neurons is disrupted, associated with disordered radial glial morphology. In vitro, SDF1α, a direct Foxc1 target also expressed in the head mesenchyme, acts as a cerebellar radial glial mitogen and a chemoattractant for nascent Purkinje cells. Its receptor, Cxcr4, is expressed in cerebellar radial glial cells and conditional *Cxcr4* ablation with *Nes-Cre* mimics the *Foxc1*[−/−] cerebellar phenotype. SDF1α also rescues the *Foxc1*[−/−] phenotype. Our data emphasizes that the head mesenchyme exerts a considerable influence on early embryonic brain development and its disruption contributes to neurodevelopmental disorders in humans.

*For correspondence: kathleen.millen@seattlechildrens.org

**Competing interests:** The authors declare that no competing interests exist.

**Reviewing editor**: Robb Krumlauf, Stowers Institute for Medical Research, United States

## Introduction

Dandy-Walker malformation, the most common congenital human cerebellar malformation, is defined by cerebellar vermis hypoplasia, an enlarged fourth ventricle and an enlarged posterior fossa (*Parisi and Dobyns, 2003*). Heterozygous loss of the *Foxc1* gene contributes to DWM. Although loss of this transcription factor causes significant developmental cerebellar pathology in humans and mice, in mice, its expression is limited to vascular pericytes within the developing cerebellum. In contrast, *Foxc1* is widely expressed in the posterior fossa mesenchyme surrounding the cerebellar anlage beginning after e12.5 in mice. We therefore hypothesized that disrupted *Foxc1*-dependent signalling from the posterior fossa head mesenchyme to the adjacent developing cerebellum is key to the DWM phenotype. We previously reported reduced mesenchymal expression of several secreted factors in the mouse Foxc1 e12.5 mutant posterior fossa including Bmp 2, 4 and SDF1α (*Aldinger et al., 2009*). However, *Foxc1* mutant cerebellar developmental abnormalities were not fully investigated and the relevance of these mesenchymal expression changes remained unexplored.

The cerebellar anlage derived from dorsal rhombomere 1 is initially established by Fgf8 and other signalling molecules originating from the isthmic organizer, a transient embryonic mid/hindbrain junction at neural tube closure (*Nakamura et al., 2005*; *Basson and Wingate, 2013*). Initial neurogenesis in the cerebellar anlage occurs in two distinct germinal zones. The cerebellar ventricular zone gives rise to waves of GABAergic neurons including cerebellar Purkinje cells which migrate radially away from the ventricular zone into the developing cerebellar cortex (*Seto et al., 2014*). The cerebellar rhombic

**eLife digest** The part of the brain responsible for coordinating and fine-tuning movement, sensory processing and some cognitive functions—the cerebellum—is found tucked away at the back of the brain, where it sits in a hollow in the skull called the posterior fossa. In a relatively common neurological disorder called Dandy-Walker malformation, part of the cerebellum doesn't develop and the posterior fossa is abnormally large.

One contributing factor to Dandy-Walker malformation is the loss of a protein called Foxc1. This protein is a so-called transcription factor, meaning it activates other genes, and so it has various important roles in helping an embryo to develop. In mouse embryos, the gene that produces Foxc1 is not activated in the developing cerebellum itself, but rather in the adjacent mesenchyme, a primitive embryonic tissue that will develop into the membranes that cover the brain and the skull bones that define the posterior fossa. This led Haldipur et al. to propose that the mesenchyme and the cerebellum communicate with each other as they develop.

To investigate this idea, Haldipur et al. carefully analysed how the development of the mouse cerebellum goes awry when Foxc1 is absent. This revealed that Foxc1-deficient mice have lower numbers of a type of cell called radial glial cells in their cerebellum. These are 'progenitor' cells that develop into the various types of cell found in the cerebellum, and also act as a scaffold for other neurons to migrate across. Therefore, the loss of radial glial cells in Foxc1-deficient mice substantially disrupts how the cerebellum develops, and how the neurons in the cerebellum work.

One gene activated by the Foxc1 protein encodes another protein called SDF1-alpha. This protein is released from the tissue that will develop into the posterior fossa, and binds to a receptor protein that is present on radial glial cells in the cerebellum. When this binding occurs, the radial glial cells grow and divide, and so the embryo's cerebellum also grows. Haldipur et al. found that mouse embryos specifically missing this receptor develop many of the abnormalities seen in Foxc1-deficient mice and further, when SDF1-alpha was provided back into Foxc1-deficient cerebella, the defects were rescued. This suggests that the cerebellar defects caused by the loss of Foxc1 stem from disrupting the signalling pathways that are triggered by the interaction between SDF1-alpha and its receptor.

These studies highlight that the brain does not develop in isolation. It is strongly dependent on the signals it receives from the embryonic mesenchyme that surrounds it. Identifying these signals and understanding how they can be disrupted by both genetic and non-genetic causes, such as inflammation, may be key to understanding this important class of brain birth defects.

lip gives rise to cerebellar glutamatergic neurons including cerebellar granule neuron progenitors (GNPs), which migrate tangentially over the developing anlage to form the external granule layer (EGL). From the EGL, newly born granule cells migrate inward radially to form the mature internal granule layer (*Millen and Gleeson, 2008*). Extrinsic signalling has previously been implicated in some aspects of mouse cerebellar development. Transventricular Shh from the choroid plexus regulates cerebellar ventricular zone proliferation from e14.5 (*Huang et al., 2010*). Mesenchymal Bmp signals induce the cerebellar rhombic lip around e10.5 in mice (*Alder et al., 1996*, *1999*; *Fernandes et al., 2012*; *Tong and Kwan, 2013*). Additionally, meningeal SDF1α has a chemoattractive role in both the tangential migration of GNPs away from the rhombic lip to form the EGL from e12.5, as well as a later role in maintaining their proliferative niche adjacent at the pial surface within the EGL (*Hartmann et al., 1998*; *Ma et al., 1998*; *Zou et al., 1998*; *Klein et al., 2001*; *Reiss et al., 2002*; *Zhu et al., 2002*; *Vilz et al., 2005*; *Tiveron and Cremer, 2008*; *Yu et al., 2010*).

To investigate which events of cerebellar development are disrupted by loss of *Foxc1*–dependent secreted factors, we first undertook an extensive phenotypic analysis of *Foxc1*−/− embryonic cerebellar development. This analysis uncovered a dramatic deficit in cerebellar ventricular zone radial glial cell proliferation and differentiation during early cerebellar anlage development. Using a combination of in vitro and in vivo assays, we next determined that SDF1α, a direct transcriptional target of Foxc1 (*Zarbalis et al., 2012*) is a major effector of Foxc1 cerebellar pathogenesis. SDF1α secreted by the posterior fossa mesenchyme acts through Cxcr4 in cerebellar radial glial cells to drive their

proliferation and hence embryonic cerebellar anlage growth. We also show that SDF1α−Cxcr4 signalling is required to maintain the radial glial scaffold for ventricular zone derived neuronal migration. These studies dramatically expand the known roles of SDF1α-Cxcr4 signalling to encompass almost every major developmental program of cerebellar development.

## Results

### Loss of mesenchymal *Foxc1* reduces proliferation and increases differentiation in the embryonic cerebellar ventricular zone

*Foxc1* expression is initiated in the head mesenchyme adjacent to the developing cerebellum at e12.5. Although the *Foxc1*⁻/⁻ mutant cerebellum is not dramatically affected at e12.5, by e18.5 the mutant cerebellum is highly dysmorphic (*Aldinger et al., 2009*). To better define the effect of *Foxc1* deletion on pre-natal cerebellar development, we assessed cerebellar morphology at e13.5, e15.5 and e17.5 (*Figure 1A–H*). Cresyl violet staining confirmed dramatic rhombic lip and EGL abnormalities (*Aldinger et al., 2009*) at all stages, however we also noted extensive additional abnormalities of the cerebellar ventricular zone and its derivatives in mutant animals as early at e13.5, when cerebellar anlage size and shape differences were readily apparent (*Figure 1D*). Unlike the forebrain mesenchyme, the posterior fossa mesenchyme in the *Foxc1*⁻/⁻ mutant was structurally normal, as indicated by Laminin, Pdgfr1 and Raldh2 staining (*Figure 1—figure supplement 1*).

Since cerebellar anlage size during early development is directly related to cerebellar ventricular zone proliferation, we assessed proliferation by sacrificing animals immediately following a 1 hr BrdU pulse at e13.5 and e15.5. The VZ was identified as a layer of cells directly lining the fourth ventricle. Ki67 staining confirmed the boundary of the VZ (*Figure 1—figure supplement 2*). In wild-type animals, proliferation is normally downregulated across these time points as the ventricular zone becomes depleted (*Sudarov et al., 2011*) (Compare *Figure 1I* to *Figure 1K*). Dramatically, at e13.5, *Foxc1*⁻/⁻ mutants demonstrated a twofold reduction of ventricular zone proliferation, compared to e13.5 wild-type animals. At e15.5, ventricular zone proliferation was also significantly lower in *Foxc1*⁻/⁻ mutants compared to control animals (*Figure 1I–M*).

Reduced proliferation in *Foxc1*⁻/⁻ mutants was accompanied by a concomitant substantial increase in differentiation as visualized by β-III Tubulin expression, an early marker of differentiating neurons (*Brazelton et al., 2000*). This was readily apparent in *Foxc1*⁻/⁻ mutants at e13.5 with extensive differentiating neurons distributed throughout the mutant anlage at this early stage (*Figure 1O*). In contrast, wild-type animals at a comparable stage exhibited β-III Tubulin antibody staining almost entirely restricted to the nuclear transitory zone containing differentiating RL derived neurons (*Figure 1N*, NTZ). In e15.5 wild-type animals, the β-III Tubulin positive cells were widely distributed within the developing anlage, however a β-III Tubulin negative region consisting of two to three layers of cells defined the ventricular zone at this stage (*Figure 1P,P'*). In striking contrast, the mutant ventricular zone was occupied by numerous β-III Tubulin–positive cells in e15.5 *Foxc1*⁻/⁻ mutant animals (*Figure 1Q,Q'*). Together, these data demonstrate that loss of Foxc1 does not just disrupt the cerebellar rhombic lip which is adjacent to the head mesenchyme where Foxc1 is normally expressed. Rather, loss of Foxc1 profoundly disrupts all early cerebellar progenitor zones.

### Abnormal Purkinje cell migration in *Foxc1*⁻/⁻ mutant embryonic cerebellum

Cerebellar Purkinje cells are generated between e10.5 and e13.5 (*Sudarov et al., 2011*) and migrate out of the ventricular zone beginning at e14.5. At e15.5 in wild-type animals, we observed Purkinje cells defined by Calbindin-expression in a domain overlying the ventricular zone (*Figure 2A,A'*, asterisk). By e19.5 these wild-type cells had migrated outwards to form a multilayered Purkinje cell plate underneath the external granule layer in the developing cerebellar cortex (*Figure 2C,C'*, asterisk). As expected at e15.5 in *Foxc1*⁻/⁻ mutants, fewer Purkinje cells were present since proliferation of the ventricular zone was compromised during the peak of Purkinje cell production. Purkinje cell migration however was also affected. While some cells remained in the ventricular zone at this stage, most remaining calbindin expressing cells were scattered throughout the anlage (*Figure 2B,B'*, asterisk). By e19.5 the majority of the mutant Purkinje cells remained within the core of the cerebellar anlage with just a few present in a residual dysmorphic Purkinje cell plate located in the anterior cerebellum (*Figure 2D,D'*, asterisk).

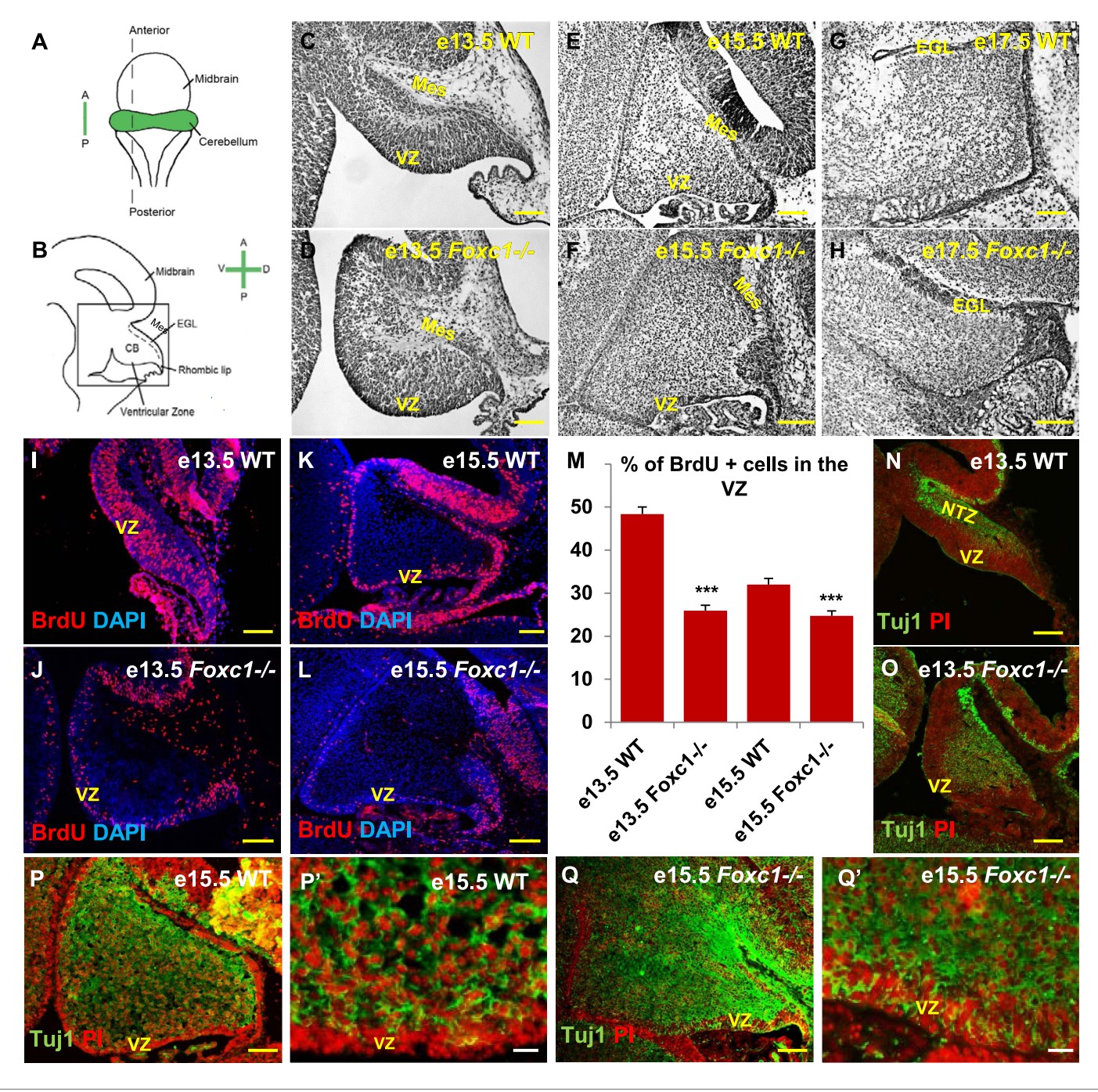

**Figure 1**. *Foxc1* deletion leads to reduced proliferation and increased differentiation in the cerebellar ventricular zone. (**A**) Schematic of a dorsal whole mount view of the embryonic mouse brain and (**B**) Sagittal section of the mid-hindbrain region. Abbreviations include the four axes; A—Anterior, P—Posterior, D—Dorsal, and V—Ventral; CB—Cerebellum and EGL—External Granule Layer. (**C–H**) Sagittal sections of e13.5 (**C**, **D**), e15.5 (**E**, **F**) and e17.5 (**G**, **H**) cerebellum from WT (**C**, **E**, **G**) and *Foxc1$^{-/-}$* (**D**, **F**, **H**) mice. (**I–L**) *Foxc1$^{-/-}$* mice showed a reduction in proliferation at e13.5 (**J**) and e15.5 (**L**), compared to WT (**I**, **K**). (**M**) Graph showing the percentage of BrdU positive cells in the cerebellar ventricular zone. The percentage of BrdU positive cells in the ventricular zone of the *Foxc1$^{-/-}$* cerebellum was significantly lower than WT at e13.5 and e15.5. Data represented as mean percentage of BrdU positive cells ± s.e.m. *** indicates significance with respect to the corresponding WT Control (p < 0.05). (**N–Q'**) β-III Tubulin staining of the e13.5 (**N–O**) and e15.5 (**P–Q'**) cerebellum showed by increased differentiation in the ventricular zone of *Foxc1* mutants at both e13.5 (**O**) and e15.5 (**Q**, **Q'**) compared to WT (**N**, **P–P'**). Abbreviations used; Mes—Mesenchyme, VZ—Ventricular Zone. Scale bar = 100 μm for all images, except **P'** and **Q'** where the scale bar = 20 μm.

*Figure 1. Continued on next page*

*Figure 1. Continued*

The following figure supplements are available for figure 1:

**Figure supplement 1**. Posterior fossa meningeal formation is not compromised in the *Foxc1⁻/⁻* mutants.

**Figure supplement 2**. Ki67 expression in the *Foxc1⁻/⁻* and Cxcr4 conditional knockout cerebellum.

### Radial glia and Bergmann glia are disrupted in the *Foxc1⁻/⁻* mutant cerebellar anlage

Since we observed dramatic Purkinje cell migration defects, we next examined the status of radial glial cells (*Figure 3A–H*) and Bergmann glia (*Figure 3J–M*). RG cells serve as both neuronal progenitors (*Feng et al., 1994*) and a scaffold for Purkinje cell neurons as they migrate towards the cerebellar cortex (*Yuasa et al., 1996*; *Hatten, 1999*) while BG serve as a scaffold for migrating granule neurons from the EGL. In wild-type animals at e13.5 and e15.5, the nestin-positive soma of RG were located in the ventricular zone with radial fibers extending to the NTZ or pial surface (*Figure 3A*, arrows). By e15.5, plentiful nestin-positive radial glial fibers extending from the ventricular surface to the pial surface were apparent in the wild-type anterior cerebellar anlage (*Figure 3C–C'*, arrows). However, by e17.5, radial glial cells in the posterior wild-type anlage transitioned into Bergmann Glial cells with radial fibers spanning just the presumptive molecular layer in the developing cerebellar cortex (*Li et al., 2014*) (*Figure 3E,G*; arrows). In *Foxc1⁻/⁻* mutants at e13.5, nestin-positive staining was evident; however extended radial fibers were absent (*Figure 3B*). By e15.5 nestin-positive fibers were discontinuous with very few extending to the pial surface (*Figure 3D,D'*, arrows). A similar absence of long radial fibers was evident in the e17.5 mutant (*Figure 3F*, arrows). Additionally, Bergmann glial morphology and arrangement were found to be disrupted in the *Foxc1⁻/⁻* mutant (*Figure 3F–H,J–M*,

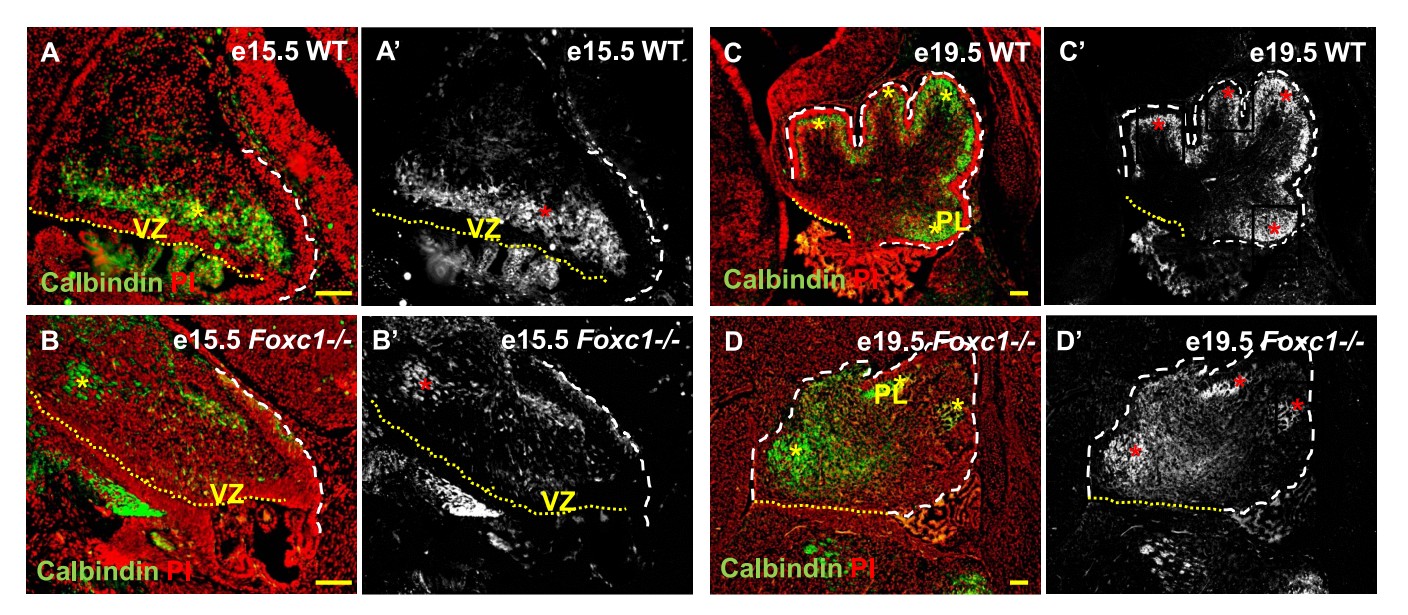

**Figure 2**. Abnormally positioned Purkinje cells are present in the cerebellum of *Foxc1 −/−* mice. (**A–D**) Sagittal sections of the WT (**A**, **A'**; **C**, **C'**) and *Foxc1⁻/⁻* cerebellum (**B**, **B'**, **D**, **D'**) stained for Calbindin. In the e15.5 WT cerebellum (**A–A'**, asterisk), Purkinje cells were present in a band overlying the ventricular zone. However, in the *Foxc1* null mutant, (**B–B'**, asterisk) fewer ectopically located PCs were present. By e19.5, PCs in the WT (**C–C'**) formed a layer of cells directly underneath the ML in the cerebellar cortex (**C'**, asterisk). However, in the *Foxc1* null mutant (**D**, **D'**), the PCs were arranged as clusters within the anlage (**D'**, asterisk). Abbreviations used; PL—Purkinje Layer and VZ—Ventricular Zone. The white dashed line indicates the outer boundary of the EGL and the mesenchyme, while the yellow dotted line represents the cerebellar ventricular surface. The red and yellow asterisks represent the PL. **A'–D'** are grey scale images of A-D respectively. Scale bar = 100 μm.

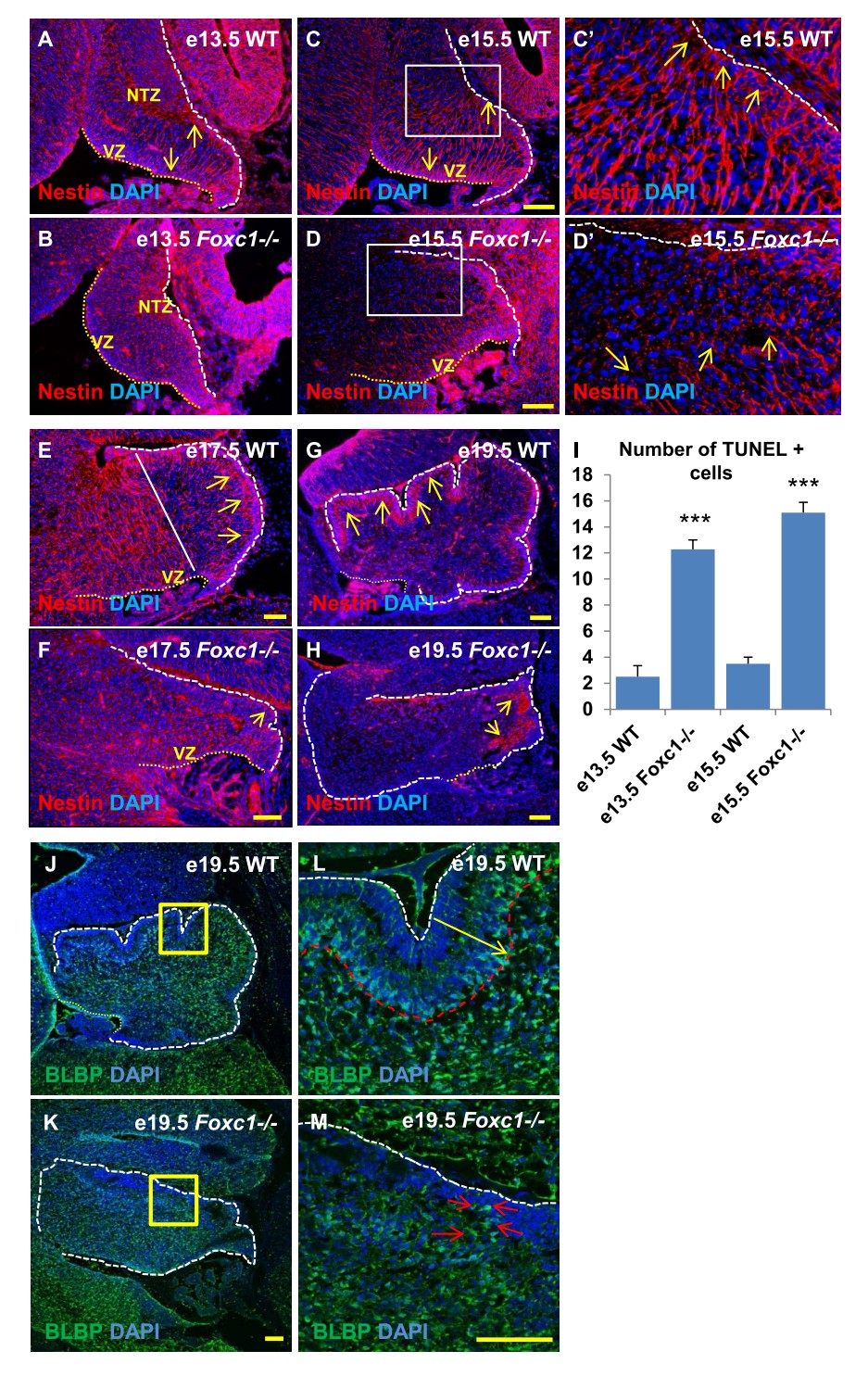

**Figure 3**. Radial glial and Bergmann glial morphology is severely disrupted in *Foxc1−/− mice*. (**A–H**) Sagittal sections of the embryonic mouse cerebellum in WT (**A**, **C**, **C'**, **E**, **G**) and *Foxc1* null mutants (**B**, **D**, **D'**, **F**, **H**) at e13.5 (**A**, **B**), e15.5 (**C–D'**), e17.5 (**E**, **F**) and 19.5 (**G**, **H**) stained for Nestin. While radial glial fibers extended from the ventricular zone to the pial surface in the WT cerebellum (**C**, **C'**, arrows), in the *Foxc1* null mutant, fibers were discontinuous and did not extend all the way to the pial surface (**D**, **D'**, arrows). In the WT cerebellum, Bergmann glial fibers extended from the EGL to the PL at e17.5 (**E**) and e19.5 (**G**, **L**; arrows). The white straight line in (**E**) demarcates the anlage into two regions—an anterior (left) region where fibers extend from the VZ to the pia, and the
*Figure 3. Continued on next page*

*Figure 3. Continued*

other (right) where Bergmann glial fibers extend from the EGL to the IGL. In the *Foxc1* null mutant, these two zones were not apparent and fiber morphology was severely disrupted (**F**, **H**; arrows). (**I**) Graph showing the number of TUNEL positive cells that span the length of the cerebellar ventricular zone. The number of TUNEL positive cells in the ventricular zone of the *Foxc1*⁻/⁻ cerebellum was significantly higher than WT at e13.5 and e15.5. Data represented as average number of TUNEL positive cells per section analysed ± s.e.m. *** indicates significance with respect to corresponding WT Control (p < 0.05). (**J–M**) Sagittal sections of the WT (**J**, **L**) and *Foxc1*⁻/⁻ cerebellum (**K**, **M**) stained for BLBP. In the e19.5 WT cerebellum (**J**, box; **L**, arrow), BG were present as a layer overlapping with the PL. However, in the *Foxc1* null mutant, (**K**, box; **M**, arrow) BG were ectopically located in the EGL. Abbreviations used; NTZ—Nuclear Transitory Zone and VZ—Ventricular Zone. The white dashed line indicates the outer boundary of the EGL and the mesenchyme, while the yellow dotted line represents the cerebellar ventricular surface. The white straight line in Image E represents the boundary between two zones—one where fibers extend from the VZ to the pia, and the other where BG fibers extend from the EGL to the IGL. **C′**–**D′** are high magnification images of **C**–**D** respectively. Scale bar = 100 μm.

*Figure 3K,M*; arrows). In the WT, BG were found to overlap with the PL (*Figure 3E,G,J,L*), while in the *Foxc1*⁻/⁻ mutant cells were found to be present ectopically in the EGL and ML indicating abnormal migration (*Figure 3K,M*). These changes were accompanied by increased cell death in the ventricular zone of *Foxc1*⁻/⁻ mutants as measured by TUNEL labeling (*Figure 3I*). Together, our results suggest that the disruption of radial glial migratory scaffold in *Foxc1*⁻/⁻ mutants contributes to aberrant PC migratory phenotypes in *Foxc1*⁻/⁻ mutant mice, which in turn also contribute to an abnormal BG phenotype.

## The SDF1α -Cxcr4 signalling pathway regulates cerebellar ventricular zone proliferation and differentiation in vitro

We hypothesized that Foxc1 regulates the expression of secreted factors in the head mesenchyme which then non-autonomously influence the developing cerebellum. Based on candidate gene analysis, we previously reported that *Foxc1*⁻/⁻ mutant posterior fossa mesenchyme has reduced expression of SDF1α (Cxcl12), Bmp2 and 4 (*Aldinger et al., 2009*). We tested the ability of these secreted factors to influence cerebellar ventricular zone proliferation and migration of ventricular zone-derived neurons using various assays (*Figure 4*). The embryonic cerebellar ventricular zone does not express Bmp-receptors although they are expressed in the rhombic lip (*Machold et al., 2007*). In contrast, the SDF1α receptor, Cxcr4 is highly expressed in the embryonic ventricular zone (*Figure 5* and data not shown). Addition of SDF1α significantly increased cell division in primary dissociated cultures of the wild-type e13.5 cerebellar anlage. Blocking SDF1α function using the Cxcr4 receptor antagonist AMD 3100 (*Rosenkilde et al., 2004*), significantly inhibited BrdU uptake in these cultures (*Figure 4B*). The majority of cerebellar progenitors at e13.5 derive from the cerebellar ventricular zone, although both cerebellar rhombic lip progenitors and a small number of granule cell progenitors are also present at this time. To determine if the anti-proliferative AMD3100 could specifically alter cerebellar ventricular zone proliferation, we cultured whole e13.5 cerebellar anlage explants in serum containing media with or without AMD 3100 for 24 hr. Subsequent sections of these whole mount cerebellar cultures revealed that blockage of the SDF1α-Cxcr4 signalling pathway caused a significant increase in ventricular zone neuronal differentiation and reduced proliferation. (*Figure 4A,C–F*).

SDF1α has chemoattractant properties on cerebellar EGL cells where it is required in the post-natal pia to maintain the adjacent EGL cells in their proliferative niche (*Ma et al., 1998*; *Zou et al., 1998*; *Klein et al., 2001*; *Reiss et al., 2002*). Earlier, it also acts as a chemoattractant for cells migrating from the RL to form the EGL (*Yu et al., 2010*). Since we observed Purkinje cell migration deficits in *Foxc1*⁻/⁻ mutants, we assessed if SDF1α has yet another role, acting as a pial surface chemoattractant to differentiating ventricular zone derived cells. Indeed, SDF1α significantly increased the number of e13.5 dissociated cerebellar cells that migrated through a membrane in a standard transwell migration assay. Further, addition of AMD 3100 to cells in the upper chamber significantly reduced the number of migrating cells. (*Figure 4G,H*). To confirm that the cells that were responsive to SDF1α responsive cells were indeed cells arising from the ventricular zone and not just the rhombic lip, we performed a matrigel migration assay using explanted e13.5 cerebellar anlage where ventricular zone derived cells were lineally labelled by a Td-tomato reporter (Ptf1a cre/+; Ai14) (*Figure 4I*). Labelled e13.5 explants

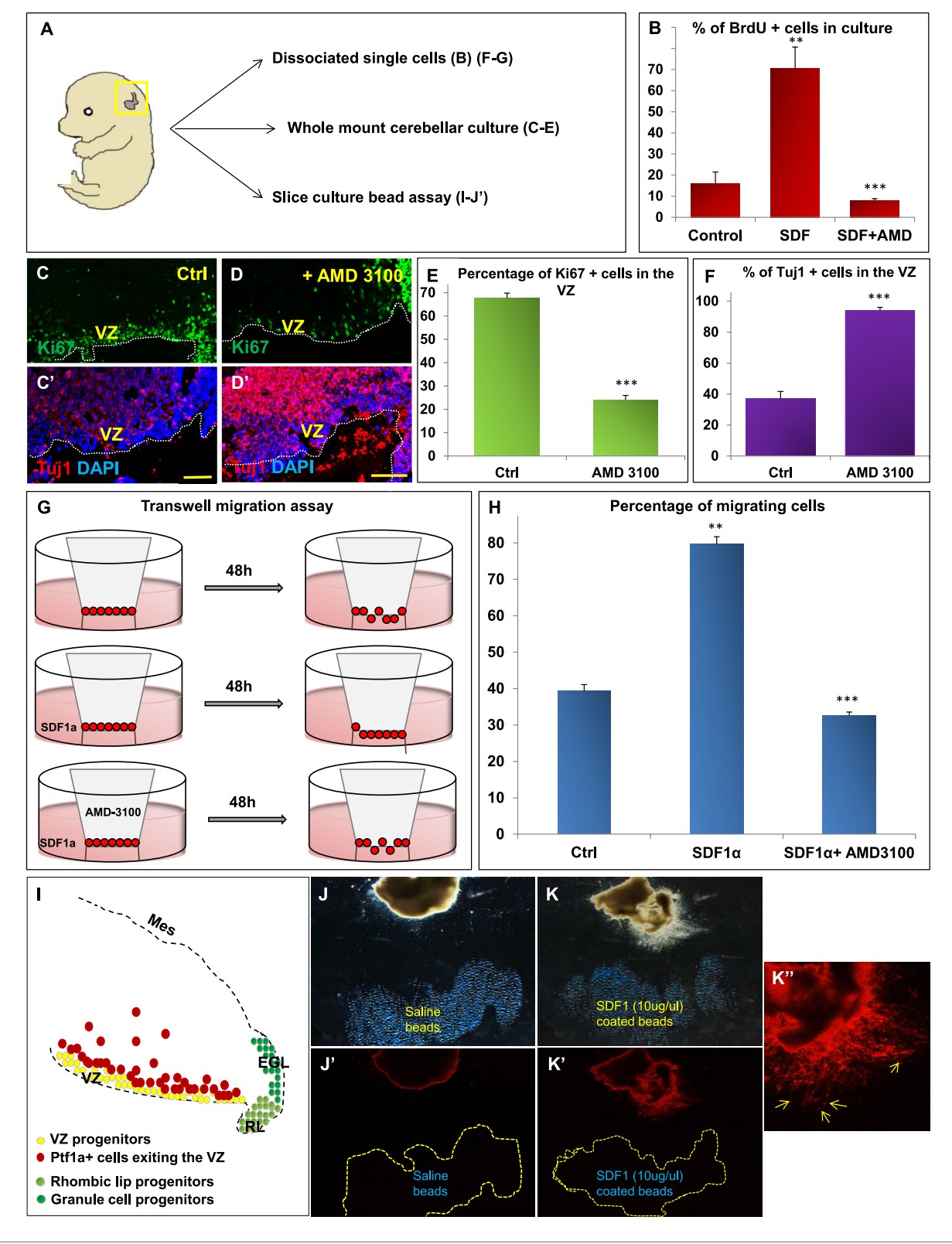

**Figure 4**. SDF1α induces cell division in cerebellar ventricular zone progenitors and also functions as a chemoattractant. (**A**) Schematic of dorsal whole mount view of the embryonic mouse head and a brief description of experiments included in the figure. (**B**) Graph showing the percentage of BrdU+ cells in primary dissociated cerebellar culture. Addition of SDF1α significantly increased BrdU uptake by cells. Addition of AMD3100, an antagonist of SDF1α

*Figure 4. Continued on next page*

*Figure 4. Continued*

significantly reduced BrdU incorporation. (**C**, **D**; **C'**, **D'**) Sagittal sections of e13.5 cerebellum cultured whole mount for 1 day and stained for Ki67 (inset) and β-III Tubulin. Addition of AMD3100 to the culture significantly reduced cell proliferation and increased differentiation in the ventricular zone (**D**). (**E**) Graph showing the percentage of Ki67 and (**F**) β-III Tubulin+ cells in the cerebellar ventricular zone. (**G**) Schematic describing the transwell migration assay. Addition of SDF1α significantly increased the number of cells migrating through the membrane of the insert. Addition of AMD3100 to the upper well significantly lowered the number of migrating cells (**H**). (**H**) Graph quantifying the results of the experiment described in (**G**) (**I**) Schematic of a sagittal section of the e13.5 cerebellum. Ventricular zone progenitors are labelled in yellow, while Ptf1a positive cells that exit the ventricular zone are marked red. RL progenitors are labelled light green while granule cell progenitors exiting the RL are labelled dark green. (**J–K''**) Matrigel assay to study the effect of SDF1α on the neuronal migration. Ptf1a positive cells from an e13.5 cerebellar slice, when incubated with SDF1α coated acrylic beads were seen to move towards the source of the chemokine (**K–K''**). Saline coated beads had no effect on migration (**J**, **J'**). Abbreviations used; VZ—Ventricular Zone, RL—Rhombic Lip, EGL—External Granular Layer and Mes—Mesenchyme. The white dotted line in (**C–D'**) represents the cerebellar ventricular surface. Scale bar = 100 μm. In Graphs (**B**), (**E**, **F**) and (**H**), data is represented as mean percentage of BrdU or β-III Tubulin positive cells or migrating cells ± s.e.m. In (**B**) and (**H**) ** indicates significance with respect to Control, while *** indicates significance with respect to SDF1α treatment. In (**E**, **F**) *** indicates significance with respect to Control. ($p < 0.05$), for all data.

were embedded in matrigel adjacent to acrylic beads coated with saline or SDF1α (*Figure 4J,K*). After 48 hr, Td-tomato positive cells were specifically directed towards the source of SDF1α (*Figure 4K,K',K''*). Thus, not only can SDF1α promote cerebellar ventricular zone progenitor proliferation, it can also function as a chemoattractant to cerebellar ventricular zone derived cells.

## Cxcr4 is expressed in the cerebellar radial glial cells

*Cxcl12*, the gene encoding SDF1α is a direct target of Foxc1 (*Zarbalis et al., 2012*). It is expressed in embryonic posterior fossa mesenchyme based on RT-PCR data and its expression is reduced in *Foxc1⁻/⁻* mutants (*Aldinger et al., 2009*). Using a BAC transgenic mouse encoding an SDF1α:mRFP fusion protein (*Jung et al., 2009*; *Mithal et al., 2013*), we confirmed that SDF1α protein is expressed in mesenchymal cells adjacent to the cerebellar anlage at e13.5. In addition, we observed scattered RFP-positive cells, likely pericytes, within the cerebellar anlage at this stage (*Figure 5A*, arrow). To extend previous reports of cerebellar ventricular zone expression of the SDF1α receptor (*Zou et al., 1998*; *Tissir et al., 2004*), we assessed Cxcr4 expression using both a BAC-transgenic Cxcr4-GFP transcriptional reporter mouse(*Mithal et al., 2013*) (*Figure 5B,C*, arrow), and Cxcr4 immunohistochemistry (*Figure 5E,F*, arrow). At e13.5, Cxcr4 was expressed around the soma of cells in the ventricular zone, the rhombic lip, the nascent EGL, the NTZ and the pia (*Figure 5B*). Strikingly, Cxcr4 expression was also seen in fibers extending out of the ventricular zone towards the pial surface. These fibers were nestin-positive and thus are radial glial fibers (*Figure 5D*, arrow). As expected of radial glial fibers, they persisted at e14.5 and became restricted to the anterior cerebellum by e16.5 (compare *Figure 5F* with *3E*). Unlike nestin however, Cxcr4 expression was not retained in Bergmann Glial cells of the developing cerebellar cortex.

## Cxcr4 conditional knockout mice exhibit cerebellar abnormalities similar to those observed in *Foxc1⁻/⁻* mutants

Both our in vitro and in vivo expression data strongly suggested that SDF1α-Cxcr4 signalling could influence both the proliferation of cerebellar ventricular zone progenitors and the migration of its derivatives, making this pathway a highly attractive downstream effector of *Foxc1* during embryonic cerebellar development. Previous phenotypic descriptions of Cxcr4 and SDF1α KO cerebellar phenotypes have focused on the role of this pathway in EGL development and maintenance (*Ma et al., 1998*; *Zou et al., 1998*). Potential earlier roles in cerebellar ventricular zone development have never been addressed. Thus, we examined the embryonic phenotypes of *Cxcr4ᶠˡ/ᶠˡ*; nestin-cre (*Cxcr4 CKO*) mice, where Cxcr4 function was specifically removed from radial glial cells and their derivatives.

As seen in *Foxc1⁻/⁻* mutant cerebella, numbers of BrdU-positive ventricular zone cells were also dramatically reduced in the Cxcr4 CKO at e13.5 and e15.5 (*Figure 6A–D,I*, Graph). Concurrently, there was an extensive increase in numbers of differentiated β-III tubulin positive cells at e13.5 and e15.5. Similar to Foxc1 null mutants, numerous β-III tubulin -positive cells were also ectopically interspersed throughout the diminished Cxcr4 CKO mutant ventricular zone at both stages (*Figure 6F,H*, Boxed area). Again, similar to *Foxc1⁻/⁻* mutant few calbindin-positive Purkinje cells were present at e15.5 in the Cxcr4 CKO and those remaining were scattered throughout the anlage and located adjacent to

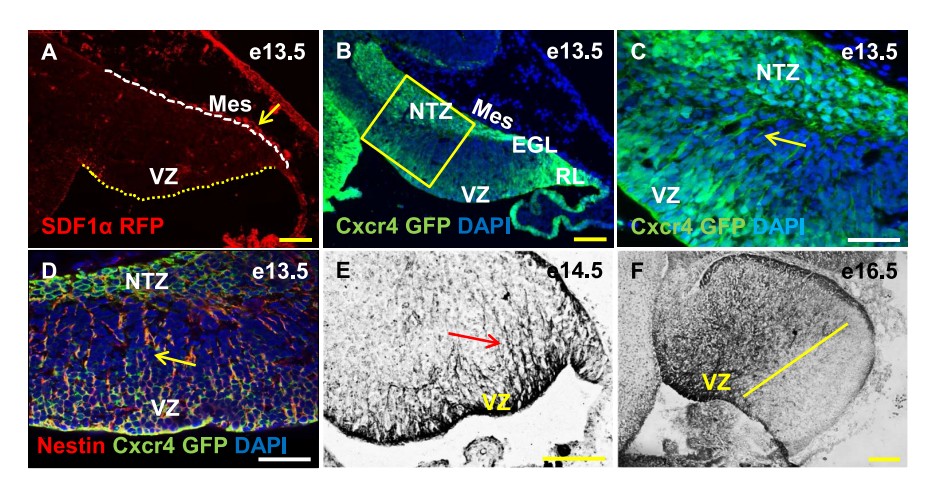

**Figure 5**. Cxcr4, a receptor of SDF1α, is strongly expressed in the soma and fibers of cerebellar radial glia. (**A**–**F**) Sagittal sections of the developing mouse cerebellum at e13.5 (**A**, **B**, **C**, **D**), e14.5 (**E**) and e17.5 (**F**). RFP-tagged SDF1α is secreted by the mesenchyme (**A**, arrow). In contract, the receptor of SDF1α, GFP-tagged Cxcr4 is strongly expressed in the ventricular zone, NTZ, EGL and RL at e13.5 (**B**–**C**). Cxcr4 expression was also seen in the fibers that extended out of the ventricular zone (**C**, **D**, **E** arrow). These fibers colabelled with Nestin (**D**, arrow) indicating they were radial glial fibers. Cxcr4 expression persisted in radial glial fibers at e17.5, but only confined to the anterior region containing radial glial fibers (**F**, left of the yellow line). Abbreviations used; VZ—Ventricular Zone, NTZ—Nuclear Transitory Zone, RL—Rhombic Lip, EGL—External Granule Layer and Mes—Mesenchyme. The white dashed line in (**A**) indicates the outer boundary of the EGL and the mesenchyme, while the yellow dotted line represents the cerebellar ventricular surface. Scale bar = 100 μm for all images except **C**, **D**, where the scale bar = 50 μm.

the pia (*Figure 6L*, arrow), instead of remaining in a domain more closely associated with the ventricular zone as in wild-type cerebella (*Figure 6K*, arrow). Although many nestin-positive radial glial cells were present in the Cxcr4-CKO mutant at e13.5 and e15.5, long radial glial fibers were not present at either stage (*Figure 6M–P*). Notably, radial glial fibers were still readily detected in the midbrain (*Figure 6N*, arrows). As seen in the *Foxc1⁻ᐟ⁻* mutant, increased cell death was also observed in the Cxcr4 CKO ventricular zone compared to control at e13.5 and e15.5 (*Figure 6J*). In contrast to the *Foxc1⁻ᐟ⁻* mutant cerebellum however, nests of ectopic proliferating granule cell progenitors were present within the *Cxcr4-Nes-Cre* mutant cerebellar anlage at this stage, as others have previously reported (*Figure 6D*, asterisk). Bergmann glia were found ectopically located in the EGL similar to the *Foxc1⁻ᐟ⁻* mutant (*Figure 6R*, arrows). We conclude that loss of *Cxcr4* in the nestin-expressing radial glial cells recapitulates multiple aspects of the *Foxc1⁻ᐟ⁻* mutant cerebellar phenotype. The striking similarities of these mutant cerebellar phenotypes strongly argues that Foxc1-dependent SDF1α secretion by the posterior fossa head mesenchyme and its reception by Cxcr4 in adjacent early cerebellar anlage radial glial cells, represent a major downstream effector pathway mediating Foxc1 Dandy-Walker cerebellar pathogenesis

## Exogenously applied SDF1α rescues the *Foxc1⁻ᐟ⁻* mutant phenotype

If SDF1α is indeed one of the major effectors of the Foxc1 dependent signalling pathway, application of SDF1α in *Foxc1⁻ᐟ⁻* mutants should be able to rescue the *Foxc1⁻ᐟ⁻* phenotype. Since there are no commercially available SDF1α or Cxcr4 agonists, or posterior fossa specific cre drivers, we carried out an in vitro rescue experiment with subdissected whole cerebellar tissue from e13.5 WT and *Foxc1⁻ᐟ⁻* embryos. The cerebellar anlage was incubated with or without SDF1α (1 μg/ml) in serum free media for 12 hr. To quantify SDF1α dependent VZ proliferation rescue, the tissue was then fixed, sectioned and stained for Ki67 and β-III Tubulin (*Figure 7A–H*). SDF1α treatment did not have any effect on the WT cerebellum (*Figure 7A,E,D,H*), but led to a significant increase in proliferation in the VZ of the *Foxc1⁻ᐟ⁻* cerebellum (*Figure 7*, Compare 7B with 7A and 7C; 7D Graph). Also, while there were no differences in differentiation levels in the WT cerebellar tissue treated with and without SDF1α

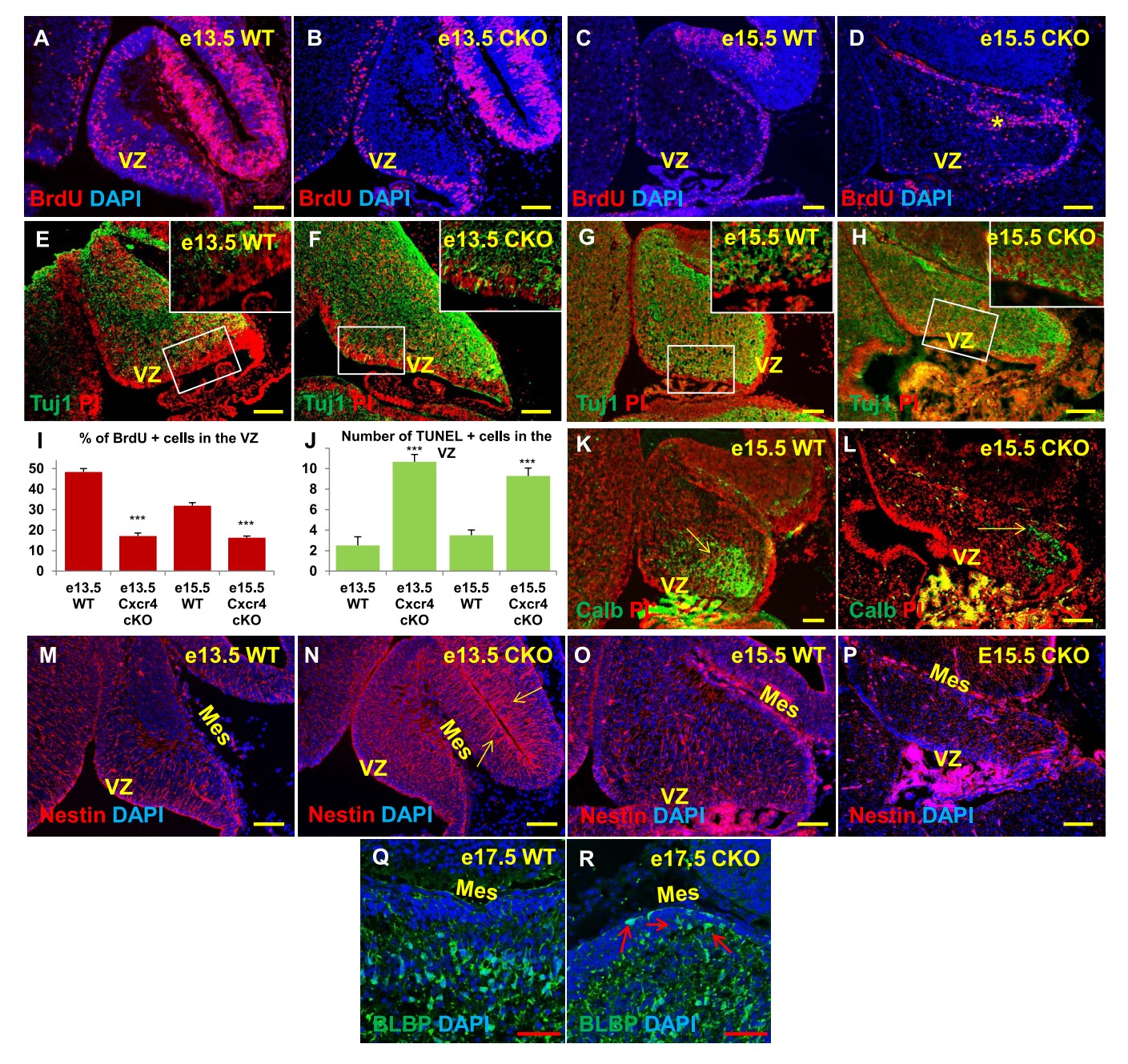

**Figure 6**. Aberrations in cell division, differentiation and migration in Cxcr4 radial glial conditional knockout mice are similar to those in Foxc1 null mutants. (**A–H**) Sagittal sections of e13.5 (**A**, **B**, **E**, **F**) and e15.5 (**C**, **D**, **G**, **H**) cerebellum from WT (**A**, **C**, **E**, **G**) and Cxcr4 Conditional Knockout (**B**, **D**, **F**, **H**) mice. (**A–D**) Cxcr4 conditional knockout mice showed a reduction in proliferation at e13.5 (**B**) and e15.5 (**D**), compared to WT (**A**, **C**). Reduced proliferation was accompanied by increased differentiation in the cerebellar ventricular zone of Cxcr4 CKO mice at both e13.5 (**F**) and e15.5 (**H**). The ventricular zone in WT cerebellum at both e13.5 (**E**) and e15.5 (**H**) largely consisted of β-III Tubulin negative cells. The insets in images (**E–H**) represent the magnified image of boxed regions within the cerebellar ventricular zone. (**I**) Graph showing a significant reduction in the percentage of BrdU positive cells in the cerebellar ventricular zone of Cxcr4 CKO compared to WT cerebellar ventricular zone. Data are represented as mean percentage of BrdU positive cells ± s.e.m. *** indicates significance with respect to corresponding WT Control. (p < 0.05). (**J**) Graph showing a significant increase in the number of TUNEL positive cells in the cerebellar ventricular zone of Cxcr4 CKO compared to WT animals. Data are represented as mean number of TUNEL positive cells ± s.e.m. *** indicates significance with respect to corresponding WT Control. (p < 0.05). (**K–L**) Sagittal sections of e15.5 cerebellum from WT (**K**) and Cxcr4 CKO (**L**) mice stained for Calbindin showing aberrant number and position of PCs in the CKO. (**K–L**) Sagittal sections of e13.5 cerebellum (**M**, **N**) and e15.5 (**O**, **P**) from WT (**M**, **O**) and Cxcr4 CKO (**N**, **P**) mice stained for Nestin. The morphology of radial glial fibers in the Cxcr4 CKO greatly resembled
*Figure 6. Continued on next page*

*Figure 6. Continued*

that in the *Foxc1* mutant. (**Q**–**R**) Sagittal sections of e17.5 cerebellum from WT (**Q**) and Cxcr4 CKO (**R**) mice stained for BLBP. The morphology and positioning of BG fibers in the Cxcr4 CKO (**R**, arrows) is similar to the *Foxc1* mutant. Abbreviations used include VZ—Ventricular Zone, and Mes—Mesenchyme. Scale bar = 100 µm for all images, except for **Q**–**R** where the scale bar = 50 µm.

(*Figure 7E,H*), SDF1α significantly inhibited the early differentiation phenotype in the *Foxc1*$^{-/-}$ cerebellum (*Figure 7*, Compare 7F with 7G and 7E; 7H Graph). These data reinforce our conclusion that SDF1α is the major downstream effector of posterior fossa Foxc1.

## Discussion

Loss of *Foxc1* is associated with Dandy-Walker malformation, the most common congenital malformation of the human cerebellum. Although extensive cerebellar pathology results from the loss of *Foxc1*, in the mouse, *Foxc1* is not highly expressed in the cerebellar anlage, but rather is expressed in the posterior fossa mesenchyme overlying the developing cerebellar anlage beginning at e12.5. This finding leads to the hypothesis that disrupted mesenchyme to brain signalling caused the *Foxc1* mutant cerebellar phenotype. We previously reported disorganization of the cerebellar RL and nascent EGL located directly adjacent to the mesenchyme by e14.5 (*Aldinger et al., 2009*). We now find that the cerebellar ventricular zone is not insulated from loss of *Foxc1* despite its distance from the posterior fossa mesenchyme. Indeed, *Foxc1* null mutants have an early and devastating reduction in ventricular zone radial glial cell proliferation, increased cell death and increased differentiation at e13.5, 1 day after failure of normal initiation of expression of *Foxc1* in the posterior fossa mesenchyme. At later stages, residual ventricular zone derived Purkinje cells in the mutant display migratory deficits which correlate with loss of radial glial fibers which normally serve as a scaffold for neuronal migration. Using in vitro and in vivo assays, we demonstrated that SDF1α, a direct target of Foxc1 in head mesenchyme (*Zarbalis et al., 2012*), via its receptor, Cxcr4 is a major effector of Foxc1-dependent mesenchymal signalling. Cxcr4 is expressed in ventricular zone progenitors, including the radial glial fibers and their endfeet which associate with the developing pia. Loss of cerebellar Cxcr4 largely phenocopies the *Foxc1* null phenotype and the *Foxc1*$^{-/-}$ phenotype can be rescued by the addition of SDF1α. Together, our data demonstrates SDF1α-Cxcr4 signalling is a major downstream effector of mesenchymal Foxc1 function and is required to maintain both radial glial proliferative and scaffolding functions in the early cerebellar anlage.

Head mesenchyme is a mixture of head mesoderm and head neural crest cells which surrounds the developing brain (*Yoshida et al., 2008*). It differentiates to form all structures between the brain and the epidermis, including the meninges which secretes the pial basement membrane the skull and the skin dermis. Signalling from the head mesenchyme and its derivatives to the developing brain has been implicated in a number of important developmental processes from brain regionalization during early neural plate and neural tube stages (*LaMantia et al., 2000*; *Le Douarin et al., 2012*; *Andoniadou and Martinez-Barbera, 2013*) through the regulation of tangential migration of Cajal Retzius cells and cortical interneurons (*Kwon et al., 2011*; *Zarbalis et al., 2012*). Physical interactions between the pial surface and the end-feet of radial glial cells are also known to be required for radial glial survival, proliferation as well as maintenance of radial fiber morphology to direct the radial migration of neurons (*Radakovits et al., 2009*; *Siegenthaler and Pleasure, 2011*). The molecular identities of these mesenchymally-derived signals are beginning to be elucidated. For example, Bmp, Fgf and Wnts are coordinately required to induce and then caudalize the neural plate and neural tube (*Andoniadou and Martinez-Barbera, 2013*). SDF1α, a chemokine has been shown to regulate tangential migration of cortical neurons. Intact integrins and laminins are required in the pial basement maintain contact with radial glial endfeet (*Radakovits et al., 2009*).

To begin to elucidate the nature of Foxc1-dependent posterior fossa head mesenchymal signalling to the developing cerebellum, we previously reported downregulation of a number of genes encoding secreted molecules in Foxc1 null posterior mesenchyme at e12.5 including Bmps 2 and 4 and SDF1α (*Aldinger et al., 2009*). SDF1α is a direct target of Foxc1 in head mesenchyme (*Zarbalis et al., 2012*). We have now demonstrated that in vitro, SDF1α could induce cerebellar progenitor proliferation and the migration of neurons; an effect which could be blocked by AMD3100, an antagonist of the Cxcr4 receptor (*Rosenkilde et al., 2004*). In vivo, published in situ data together with new protein expression

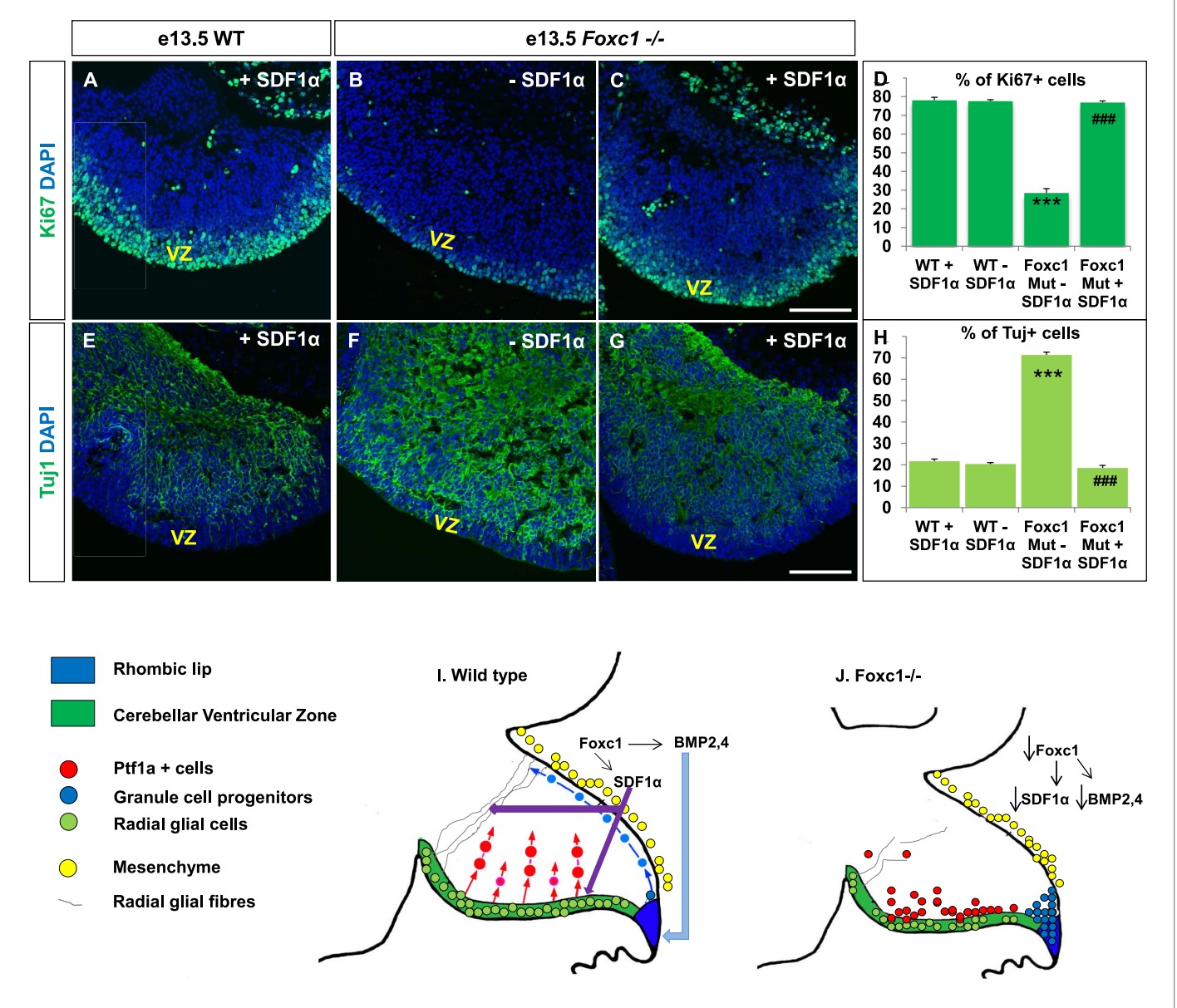

**Figure 7**. Exogenously applied SDF1α rescues the *Foxc1⁻/⁻* mutant phenotype. (**A–C**) Sagittal sections of the e13.5 whole cerebellar tissue from WT (**A**) and *Foxc1⁻/⁻* (**B–C**) embryos, stained for Ki67. Addition of SDF1α to the culture significantly increased proliferation in the *Foxc1⁻/⁻* cerebellar VZ (**C**) as compared to its mutant negative control not treated with SDF1α (**B**). Addition of SDF1α did not significantly increase proliferation in the WT ventricular zone (**A**, **D**). (**D**) Graph showing the percentage of Ki67 cells in the cerebellar ventricular zone. (**E—G**) Sagittal sections of the e13.5 whole cerebellar tissue from WT (**E**) and *Foxc1⁻/⁻* (**F–G**) embryos stained for β-III Tubulin. SDF1α significantly reduced the number of Tuj1+ cells in the VZ of the *Foxc1⁻/⁻* mutant (**G**) as compared to its mutant negative control (**F**). Addition of SDF1α did not significantly alter differentiation levels in WT embryos (**E**, **H**). (**H**) Graph showing the percentage of β-III Tubulin+ cells in the cerebellar ventricular zone. Data for both graphs (**D**) and (**H**) are represented as mean number of Ki67 or Tuj1 positive cells ± s.e.m. *** indicates significance with respect to corresponding WT Controls; ### indicates significance with respect to the corresponding *Foxc1⁻/⁻* mutant—SDF1α negative control. (**I**) Schematic of a sagittal section of the embryonic mouse cerebellum. In the WT cerebellum, mesenchymal Foxc1 controls the expression of several secreted factors such as SDF1α and Bmp2,4. SDF1α binds to its receptor Cxcr4 which is widely expressed in the RL, ventricular zone and EGL. SDF1α induces proliferation in the cerebellar ventricular zone. Not only is it required to maintain radial glia and their processes, it also functions as a chemoattractant to neurons migrating away from the ventricular zone. In the RL, Bmps maintain the progenitor pool, and SDF1α acts to attract the GC progenitors from the RL to form the EGL, thereby maintaining these progenitors at the pial surface. (**J**) In the *Foxc1⁻/⁻* cerebellum, deletion of *Foxc1* leads to a significant downregulation of SDF1α and Bmps. A reduction in SDF1α expression causes reduced ventricular zone proliferation, increased differentiation and abnormal migration. Increased cell death and dysmorphic radial glial fibers are also observed. GCs migrating out of the RL into the EGL follow abnormal paths away from the pial surface into the interior of the cerebellar anlage. Absence of this chemoattractant leads to a buildup of cells along the ventricular zone and RL.

analysis demonstrated the presence of Cxcr4 in the cerebellar ventricular zone throughout embryogenesis. In contrast, extensive in situ analysis failed to detect expression of any Bmp receptor activity in the cerebellar ventricular zone at relevant stages (*Machold et al., 2007*) (data not shown). Further, Bmp2 and 4 could not alter migration of ventricular zone derived neurons in in vitro assays at e13.5. Thus, SDF1α became a high priority candidate for further analysis.

Notably, SDF1α and Cxcr4 have previously been implicated in cerebellar granule cell development (*Ma et al., 1998*; *Zou et al., 1998*; *Klein et al., 2001*; *Reiss et al., 2002*; *Zhu et al., 2002*; *Vilz et al., 2005*; *Yu et al., 2010*). SDF1α expressed in the meninges functions as chemoattractant to direct granule neuron progenitors away from the RL to the surface of the anlage to form the EGL. In the early postnatal cerebellum, SDF1α also exerts an influence on GNPs of the outer EGL which lie directly underneath the pial surface. Loss of SDF1α or Cxcr4 in the cerebellum causes GNPs to prematurely leave their EGL proliferative niche, resulting in ectopic proliferation and dramatic disruptions of the laminar structure of the resultant cerebellum (*Ma et al., 1998*; *Zou et al., 1998*). Our results now considerably expand the role of SDF1α-Cxcr4 signalling in the developing cerebellum to include almost all aspects of radial glial function including proliferation, survival and maintenance of radial morphology (*Figure 7*).

Although the focus of previous studies of SDF1α-Cxcr4 signalling has been on neuronal migration, our new analysis now demonstrates a more fundamental role for this signalling pathway. Loss of either *Foxc1* or Cxcr4 causes dramatic 25% and 30% respective reductions in ventricular zone proliferation by e13.5 which rapidly depletes the early cerebellar progenitor pool. This difference may reflect the residual SDF1α-Cxcr4 signalling in the Foxc1 mutants (*Aldinger et al., 2009*), vs complete loss in the Cxcr4 CKO. Taken together with other published studies, we conclude that embryonic cerebellar anlage size is regulated by a number of extra-cerebellar signalling systems. Wnt signals derived from both the isthmic organizer (*Sato et al., 2004*; *Sato and Joyner, 2009*) and the early dorsal roof plate (*Chizhikov et al., 2006*) determine the initial size of the anlage from ~e9.0. We have now demonstrated that mesenchymal SDF1α signalling is a major driver of early anlage growth from ~e12.5. Late embryonic anlage ventricular zone proliferation is influenced by Shh derived from the differentiating choroid plexus from e14.5 (*Huang et al., 2010*). An outstanding issue remains how each of these signalling systems are coordinately regulated and integrated across embryogenesis to orchestrate normal anlage development.

In addition to serving as neuronal and glial progenitors in the ventricular zone, radial glial cells extend fibers that act as migratory scaffolds for neurons migrating out of the ventricular zone. We observed aberrant migration of Purkinje cells in both Foxc1 null and Cxcr4; nestin-cre mutants, in addition to dysmorphic radial glial fibers. Abnormal migration of cells out of the ventricular zone could be due to the absence or downregulation of SDF1α chemoattraction from the mesenchyme and derived meninges. This is buttressed by the fact that blockage of SDF1α led to a significant reduction in migration of single cells in the in vitro transwell migration assay. Also, in the matrigel assay, when embryonic cerebellar slices were placed in close proximity with SDF1α coated acrylic beads, explant cells and fibers migrated in the direction of the SDF1α source. However, our protein expression data using Cxcr4 antibodies, in addition to Cxcr4-GFP transgenic mice indicate that Cxcr4 is also strongly expressed in radial glial fibers themselves. As early as e13.5 and more prominently by e15.5, radial glial fibers were discontinuous and did not extend all the way to the basement membrane of the pia. We conclude that SDF1α secreted by the mesenchyme binds to the receptor on the radial fibers to relay signals that are directly required to maintain these radial fibers as a substrate for migration. Indeed in the spinal cord, SDF1α protein is detectable in radial end feet and transcytosis across the radial glial length is also observed (*Mithal et al., 2013*).

Radial glial cells in the cerebellum can differentiate into Bergmann glial cells (also known as Golgi Epithelial cells) in the cerebellar cortex (*Li et al., 2014*). In both *Foxc1*⁻ᐟ⁻ and Cxcr4 CKO mutants, Bergmann glial cells were ectopically placed in the EGL and ML, while in wild-type animals Bergman glial nestin+ and BLBP+ cells extended processes from the EGL to the presumptive IGL and were found overlapping with the PL. Studies have also shown that the association of BG with PCs greatly contributes to their structural and functional maintenance. Sonic hedgehog secretion by PCs also helps maintain BG in the cerebellum. It is hence plausible that the BG phenotype may be secondary to abnormal RG and PCs. Fate mapping studies have shown that Bergmann glial cells are normally born between e13.5 and e14.5 in the mouse cerebellum (*Sudarov et al., 2011*). It is also possible that continued SDF1α-Cxcr4 signalling is actively required for the differentiation of BG from

RG. Further analysis is required to distinguish these two possibilities. Regardless, it is certain that a lack of normal Bergmann glial fibers further contributes to the abnormal lamination in neonatal SDF1α and Cxcr4 mutant animals as these fibers are normally required for the inward migration of EGL cells to form the internal granule layer.

Our analysis indicates that radial glial cells in the developing cerebellum are more sensitive to loss of SDF1α signalling compared to other regions of the CNS. In the cerebellum, loss of SDF1α signalling causes immediate and catastrophic cell cycle exit of radial glial cells and corresponding increased neuronal differentiation. Rapid loss of radial glial fiber morphology is also observed. Loss of SDF1α signalling in the forebrain has not been reported to cause microcephaly in mice, and we have shown here that midbrain radial glial cells are relatively unaffected (*Figure 6N*). A recent study (*Mithal et al., 2013*) reported that loss of radial glial Cxcr4 in spinal cord radial glial cells causes disruption of radial glial morphology and reduced mitosis in the ventral ventricular zone by e14.5. However, in the spinal cord by this stage a large number of spinal cord neuronal populations have already been generated (*Tanabe and Jessell, 1996*). Although late born populations such as oligodendrocyte lineages are altered, there are not dramatic morphological consequences. The disproportionate mouse cerebellar phenotypes we report here with loss of Foxc1 and Cxcr4 beautifully model the CNS phenotype of Dandy-Walker malformation patients, where the cerebellum is disproportionately diminished in size and altered in morphology relative to other CNS structures. The reasons for the region-specific sensitivities to loss of the SDF1α-Cxcr4 signalling remain unknown and may simply reflect regional differences in the timing of proliferative epochs relative to the onset of SDF1α and Cxcr4 expression. Alternatively, these differences may more fundamentally reflect region specific mesenchymal signalling mechanisms that further analyses will define.

Our data contribute to a growing body of evidence that the final form and function of the brain is a product of both intrinsic and extrinsic signalling mechanisms. Here we have demonstrated that the pathogenesis underlying *Foxc1*-dependent Dandy-Walker malformation, a major human neurodevelopmental disorder, is caused by loss of SDF1α-Cxcr4 mediated cerebellar radial glial proliferative and migrational scaffold function. These functions represent a previously unrecognized role for this major signalling pathway. Thus, SDF1α function is central to almost every major developmental program of the cerebellum. Notably, SDF1α is also a chemokine, central to the body's inflammatory response (*Dotan et al., 2010*; *Werner et al., 2011*, *2013*). Maternal infection has long been recognized as a risk factor for neurodevelopmental disorders, including autism which has been associated with cerebellar hypoplasia (*Fatemi et al., 2008*; *Buehler, 2011*; *Burd et al., 2012*; *Garbett et al., 2012*). It is intriguing to speculate that neuroinflammation, including misregulated SDF1α signalling during early fetal development may underlie cases of human cerebellar hypoplasia for which genetic mechanisms have been difficult to assign (*Sajan et al., 2010*).

## Materials and methods

### Mice

All animal experimentation done in this study was done in accordance with the guidelines laid down by the Institutional Animal Care and Use Committee (IACUC), of Seattle Children's Research Institute, Seattle, WA, USA. $Foxc1^{lacZ/+}$ mice were generated by *Kume et al. (1998)* (Northwestern University). $Foxc1^{lacZ/+}$ mice were crossed and the day of plug was taken as embryonic day (e) 0.5. We refer to this allele as the *Foxc1* null allele. *Foxc1* homozygous null mutants are referred to as $Foxc1^{-/-}$ mice. Mutant and WT Control embryos were dissected out at days 13.5, 14.5, 15.5, e17.5 and e19.5 after plugging. *Nes-Cre* mice (The Jackson laboratory Stock number −003771) were crossed with $Cxcr4^{loxP}$ mice (The Jackson laboratory Stock number −008767) to generate $Nes-Cre;Cxcr4^{loxP/+}$,which were in turn crossed with $Cxcr4^{loxP}$ mice to generate *Cxcr4* CKO mice. *Cxcr4* GFP, *SDF1α* RFP mice were generated as described previously (*Jung et al., 2009*; *Mithal et al., 2013*). Embryos were fixed in 4% paraformaldeyde (PFA) overnight, washed in PBS and sunk in 30% sucrose. Embryos were subsequently embedded in optimum cutting temperature (OCT) compound. Mid-sagittal cryo-sections of 11 microns were taken. For BrdU studies, pregnant mice were injected with BrdU (100 μg/g body weight) and sacrificed 1 hr later. In order to label cerebellar ventricular zone lineage, *Ptf1a-Cre* (*Hoshino et al, 2005*) mice generated by C.V. Wright (Vanderbilt University Medical Center) were crossed with Ai14 reporter mice (*Madisen et al., 2010*) (The Jackson Laboratory Stock number—007908).

## Immunohistochemistry

Immunohistochemistry was performed as described previously (*Haldipur et al., 2011*). Sections were subjected to antigen retrieval prior to staining. All sections were blocked using 5% serum with 0.1% triton X, and then incubated with the primary antibody, overnight. The primary antibodies used in this study were BrdU (Abcam, Cambridge, UK; 1:50), β-III Tubulin (Promega, Madison, WI, USA; 1:1000), Calbindin (Swant, Switzerland; 1:3000), Cxcr4 (Abcam—1:100), BLBP (Abcam—1:100), Ki67 (Vector—1:300), Laminin (Sigma—1:25), Raldh2 (Sigma—1:400), Pdfgr1 (BD Biosciences, San Jose, CA—1:200) and Nestin (Millipore, Germany; 1:200). The following day, the following species and subtype appropriate secondary antibodies were used—biotinylated (1:250, Vector laboratories, Burlingame, CA, USA) or fluorescent dye labelled (Alexa fluors 488 and 594, 1:1000, Molecular probes, Grand Island, NY, USA). Sections were counter stained with DAPI using in Vectashield mounting media with DAPI (4',6-diamidino-2-phenylindole) (Vector laboratories) or DPX (Distyrene. Plasticizer. Xylene) mounting medium. For each antibody, one section was used as a negative control where in the section was incubated with all the above solutions except the primary antibody. For BrdU immunohistochemistry, sections were treated with 1N HCl for 10 min on ice, 2 N HCl for 30 min at RT, and washed with Borate buffer prior to blocking. TUNEL assay was carried out In Situ Cell Death Detection Kit, TMR red (Roche, Germany). Briefly, cerebellar sections were incubated in the TUNEL mix (terminal deoxynucleotidyl transferase in storage buffer and TMR red labeled-nucleotide mixture in reaction buffer) for 1 hr at RT. The slides were washed with PBS and mounted with Vectashield mounting media containing DAPI or Propidium Iodide.

## Primary and tissue culture

The cerebellar anlage from e13.5 embryos was dissected aseptically in CMF-Tyrode solution. Meninges were removed, tissue were chopped into smaller pieces and collected in CMF-Tyrode. The tissue was treated with trypsin-DNAse and then dissociated in the same solution by triturating to make a single cell suspension, pelleted and resuspended in Dulbecco's modified Eagle's medium-F-12 (DMEM-F-12) containing 15 mM HEPES, L-glutamine, pyroxidine hydrochloride (Invitrogen, Grand Island, NY, USA), N2 supplement (Invitrogen), 10% fetal calf serum, 25 mM KCl, and penicillin-streptomycin. Cells were plated onto poly-D-Lysine -coated Labtek chamber (Nunc, Roskilde, Denmark) or Poly-D Lysine coated transwell migration inserts. The cells were then cultured in serum free medium for another 24 hr, during which they were treated with the following factors—SDF1α (R and D systems, USA, 1 µg/ml), and/or AMD 3100 (5 µg/ml). The treatment continued for the next 24 hr after which cells were washed with 1× PBS and then fixed in 4% PFA. For BrdU labeling experiments, BrdU was added to serum free media at 10 µM. For whole mount cerebellar explants, the cerebellar anlage from e13.5 embryos was collected and incubated in serum containing media with or without AMD3100 for 24 hr. For the rescue experiments, the cerebellar anlage was incubated in serum free media with or without SDF1α (1 µg/ml), for 12 hr. The tissue was then fixed and processed for IHC as stated previously.

## Transwell migration

Cells from an e13.5 cerebellum were dissociated and seeded into a cell culture insert. The insert was then placed into a well of a 24 well plate supplemented with Serum Free Media and SDF1α (1 µg/ml) was added to the lower well. To test the effect of SDF1α blockage on migration, AMD3100 (5 µg/ml) was added into the insert while SDF1α was added to the media in the lower well. Cells added to the upper chamber were allowed to migrate for 72 hr. Migrated cells attached to the bottom surface of the insert were quantified by DAPI staining.

## Matrigel assay

For matrigel migration experiments, e13.5 embryos from Ptf1a Cre; Ai14 mice were collected. The cerebellar anlage was subdissected and sliced. Slices were embedded in matrigel and co-incubated with SDF1α (1 µg/ml) coated acrylic beads for 48–72 hr. Control slices were incubated with saline coated beads.

## Microscopy and image acquisition

All images were captured at room temperature. Apart from minor adjustment of contrast and brightness no additional image alteration was done. All images were captured using the Zeiss Axioimager Z1 Microscope equipped with an Axiocam MRC camera and Axiovision Rel 4.8 software (Zeiss).

## Cell counts and data analyses

All cell counts were performed using Image J software (National Institutes of Health, Bethesda, MD, USA). Our analyses include data from n > 3 and N > 5, where n refers to the total number of embryos used and N represents the total number of sections per animal. All embryos were sectioned at 11 microns sagittally. Sections between WT and mutant were carefully chosen to represent the same orientation along the mediolateral axis. To minimize bias, blind counts were performed. To evaluate proliferation and differentiation, the total number of Ki67, BrdU and Tuj1 positive cells in the ventricular zone was counted. This was followed by a total DAPI count that represented the total cell count in the ventricular zone. The percentage of proliferating and differentiating cells in the ventricular zone was represented in the graph. For cell death, the total number of TUNEL positive cells that spanned the entire length of the ventricular zone was counted. Data are represented as mean ± s.e.m. Statistical significance was determined by two tailed t-test. $p < 0.05$ was considered statistically significant.

## Acknowledgements

We thank Dr Tom Kume (Northwestern University, Chicago, IL) and Dr CV Wright (Vanderbilt University Medical Center, Nashville, TN) for providing us with the Foxc1 and Ptf1a Cre mouse strains. We would also like to gratefully acknowledge Dr Julie Siegenthaler (University of Colorado, Denver) for gifting us the Raldh2 and Pdgfr1 antibodies. We would also like to thank Cindy Wang, Jonathan Skibo and Lucas M Smith for their technical assistance; Dr Theresa A Zwingman and other members of the Millen lab for critically reading the manuscript and their valuable feedback on this study. The work described herein was supported by National Institutes of Health R01NS072441 and R01NS080390 to KJM. All co-authors have seen and agreed to the contents of this manuscript. None of the co-authors have any potential financial interests or conflict of interest with respect to this manuscript.

## Additional information

### Funding

| Funder | Grant reference number | Author |
| --- | --- | --- |
| National Institutes of Health (NIH) | R01NS072441 | Kathleen J Millen |
| National Institutes of Health (NIH) | R01NS080390 | Kathleen J Millen |

The funders had no role in study design, data collection and interpretation, or the decision to submit the work for publication.

### Author contributions

PH, Conception and design, Acquisition of data, Analysis and interpretation of data, Drafting or revising the article, Contributed unpublished essential data or reagents; GSG, OKJ, Acquisition of data, Contributed unpublished essential data or reagents; VVC, Assisted with in vitro whole mount cerebellar cultures; DSM, RJM, Generated embryos from SDF1: mRFP;Cxcr4-GFP mice; KJM, Conception and design, Analysis and interpretation of data, Drafting or revising the article

### Ethics

Animal experimentation: All animal experimentation done in this study was done in accordance with the guidelines laid down by the Institutional Animal Care and Use Committee (IACUC), of Seattle Children's Research Institute (protocol# 14208), Seattle, WA, USA.

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
