## [Decision Letter]

Thank you for sending your work entitled “Mesenchymal *FoxC1* acts through SDF1α-*Cxcr4* signaling in radial glial cells to drive embryonic cerebellar growth” for consideration at *eLife.* Your article has been favorably evaluated by a Senior editor and 3 reviewers, one of whom is a member of our Board of Reviewing Editors.

The Reviewing editor and the other reviewers discussed their comments before we reached this decision, and the Reviewing editor has assembled the following comments to help you prepare a revised submission.

In this manuscript, the authors proposed a novel model in which *FoxC1* regulates cerebellar ventricular zone (VZ) cell proliferation through meninges-derived Sdf1a acting on *Cxcr4* expressed in radial glial cells. The present experiments have the potential to provide new insights into mechanisms underlying the pathogenesis of Dandy-Walker Syndrome. Therefore, this is a potentially interesting study worthy of consideration at *eLife*, but in its current form, the data do not unambiguously support their model. A number of changes and additional data are needed to address concerns and conclusions to present a coherent and compelling case. The specific comments as follows:

1) The authors suggest that there is a significant reduction in cell proliferation in the cerebellar ventricular zone in *FoxC1-KO and Cxcr4-CKO* mutants. However, BrdU staining apparently did not work properly on the mutant sample. For example, BrdU+ cells are almost completely absent from the posterior midbrain of the *FoxC1-KO* embryo (Figure 1). It is also unclear how do the authors define a ventricular zone and count BrdU+ cells within it. The authors need to provide new data of high quality of BrdU staining. A molecular marker, such as Sox2 or Ki67, should be used to demarcate the progenitor cells in the ventricular zone.

2) Cell proliferation and cell differentiation in the ventricular zone of the cerebellar anlage differ greatly along the mediolateral axis. The sections in Figure 1 appear from very different mediolateral positions. New data with comparable sections should be provided to support their conclusion.

3) Purkinje cells are generated between E10.5 and E13.5. If neurogenesis is accelerated at E13.5 as suggested by the authors, it is unexpected that there is such a dramatic reduction in the number of Purkinje cells in *FoxC1* and *Cxcr4* mutant embryos at E15.5 (we may actually expect more Purkinje cells). It is also hard to image that lack of SDF1a-*Cxcr4* signaling would cause cell death of postmitotic Purkinje cells as *Cxcr4* is only expressed by radial glia. Furthermore, the two published studies on *Cxcr4*-deficient mice show abnormal arrangement of Purkinje cells, but no indication of reduction of Purkinje cells. In fact, the paper by the Springer group shows plentiful Calbindin+ cells in the cerebellar cortex in *Cxcr4-KO* mice. The authors should provide a satisfying explanation for the apparent loss of Purkinje cells in *FoxC1* and *Cxcr4* mutant mice, and the discrepancy from the previous reports.

4) The conclusion on the loss of Bergmann glial cells in *FoxC1-KO* is not convincing. Additional data using markers, such as BLBP and Sox9, should be provided to argument this conclusion. It is worth to note that the Littman group has shown that Bergmann glial cells are formed in the *Cxcr4*-KO cerebellum. The authors should also provide data on Bergmann glial cells in the *Cxcr4*-CKO mutants. The authors state that *Cxcr4* is absent from developing Bergmann glia (where is the data?). Is it possible that *FoxC1* regulates Bergmann glial cells through a *Cxcr4*-independent pathway?

5) The authors convincingly showed that cerebellar VZ development in *FoxC1a* and *Cxcr4* mutants are grossly abnormal, with reduced number of proliferating cells and disorganized radial glial fibers. While this observation supports the model, it does not necessarily prove that the *FoxC1* VZ phenotype is caused by demised Sdf1a expression in the meninges. One complication is that *FoxC1* is required for development of meninges in the forebrain where it regulates production of RA for proper radial glial cell development (Cell, 139: 597, 2009). If development of cerebellar meninges is similarly disrupted, then the authors need to consider the possibility that VZ phenotype in *FoxC1* mutants is caused by other factors in the meninges that are independent of Sdf1a.

6) In the *Cxcr4* mutants, an interesting possibility is that Sdf1a secreted from other sources including the choroid plexus may activate *Cxcr4* in the VZ. Therefore, the authors need to discuss this possibility and carefully examine the development of meninges and expression of Sdf1a in meninges of *FoxC1* mutants. If meningeal development is compromised as in the forebrain, the authors need to perform *FoxC1* rescue experiment (using exogenously supplied Sdf1a) to clearly support their model.

[Editors’ note: the revised article was rejected after discussions between the reviewers, but a further resubmission was accepted after an appeal against the decision.]

Thank you for choosing to send your work entitled “*FoxC1* dependent mesenchymal signalling drives embryonic cerebellar growth” for consideration at *eLife.* Your full submission of the revised manuscript has been evaluated by a Senior editor and 2 peer reviewers, plus a member of our Board of Reviewing Editors, and the decision was reached after discussions between the reviewers. Based on our discussions, we regret to inform you that your work will not be considered further for publication in *eLife*.

The reviewers feel that the revisions you have made have in part improved the manuscript. One reviewer was previously concerned about whether *FoxC1-KO* and *Cxcr4-KO* mutants have the same phenotype, including cell proliferation, Bergmann glia and Purkinje cell formation. The new data provided in the revision has partially addressed this concern. However both of the reviewers and the BRE feel that data relating to the meninges is lacking and severely reduces enthusiasm for the study. Although it is formally possible that *FoxC1-KO* disrupts other signaling molecules in the meninges, RA is an unlikely candidate. In the forebrain, loss of RA signaling due to *FoxC1-KO* blocks neurogenesis, which is very different from what is found in the cerebellum. As raised in the initial review, it is important to show normal formation of the meninges and the new data does not address this issue. This would require analysis with meninges markers. The reviewers are not convinced that the *FoxC1* phenotype is caused by Sdf1 and whether Sdf1 expression is indeed regulated by *FoxC1*. There is no data confirming that Sdf1 expression is abolished in *FoxC1* meninges. The cerebellum phenotype in *Cxcr4* mutants seems to be similar to that of *FoxC1* mutants. However, the *FoxC1* mutant cerebellum is much more severe, with the basement membrane clearly detached from the pial surface (Figure 1), which would invariably disrupt Bergmann glial end feet attachment, likely causing their mis-localization. In summary, there is concern over major conclusions without a clear indication that meninges development is not compromised. In the absence of this evidence, the study cannot rule out the involvement of other factors derived from the meninges; hence the recommendation to reject the study at this stage.

---

## [Author Response]

*1) The authors suggest that there is a significant reduction in cell proliferation in the cerebellar ventricular zone in FoxC1-KO and Cxcr4-CKO mutants. However, BrdU staining apparently did not work properly on the mutant sample. For example, BrdU+ cells are almost completely absent from the posterior midbrain of the FoxC1-KO embryo (*Figure 1*). It is also unclear how do the authors define a ventricular zone and count BrdU+ cells within it. The authors need to provide new data of high quality of BrdU staining. A molecular marker, such as Sox2 or Ki67, should be used to demarcate the progenitor cells in the ventricular zone*.

We have replaced images 1J and 1L with higher quality images. We report an overall reduction in BrdU incorporation in the VZ of *FoxC1* mutants. The VZ was identified as a layer of cells directly lining the 4^th^ ventricle. Ki67 staining was done to better define and demarcate the VZ (Figure 1—figure supplement 1).

*2) Cell proliferation and cell differentiation in the ventricular zone of the cerebellar anlage differ greatly along the mediolateral axis. The sections in*
Figure 1
*and O appear from very different mediolateral positions. New data with comparable sections should be provided to support their conclusion*.

The orientation of the WT section was such that the isthmus and aqueduct were not visible in the original image. The sections have been reimaged confirming they are at comparable levels along the mediolateral axis (Figure 1). Note that the mutant does have a different morphology from WT.

*3) Purkinje cells are generated between E10.5 and E13.5. If neurogenesis is accelerated at E13.5 as suggested by the authors, it is unexpected that there is such a dramatic reduction in the number of Purkinje cells in* FoxC1 *and* Cxcr4 *mutant embryos at E15.5 (we may actually expect more Purkinje cells). It is also hard to image that lack of SDF1a-*Cxcr4 *signaling would cause cell death of postmitotic Purkinje cells as* Cxcr4 *is only expressed by radial glia. Furthermore, the two published studies on* Cxcr4*-deficient mice show abnormal arrangement of Purkinje cells, but no indication of reduction of Purkinje cells. In fact, the paper by the Springer group shows plentiful Calbindin+ cells in the cerebellar cortex in* Cxcr4*-KO mice. The authors should provide a satisfying explanation for the apparent loss of Purkinje cells in* FoxC1 *and* Cxcr4 *mutant mice, and the discrepancy from the previous reports*.

There is no discrepancy between our observations and previously reported data. The only study to have reported the arrangement of PCs in the embryonic *Cxcr4* mutant brain is from the Springer group (Ma et al., PNAS, 1998). Figure 5 in their paper shows the presence of PCs in the cerebellum. Although they have not quantified their data, the number of PCs is clearly reduced and they are also abnormally distributed, similar to what we have reported in the *FoxC1* and *Cxcr4* mutants. We do not state that neurogenesis is accelerated at e13.5. Our preferred explanation is that there is a depleted progenitor pool because of early differentiation onset.

*4) The conclusion on the loss of Bergmann glial cells in* FoxC1-*KO is not convincing. Additional data using markers, such as BLBP and Sox9, should be provided to argument this conclusion. It is worth to note that the Littman group has shown that Bergmann glial cells are formed in the* Cxcr4-*KO cerebellum. The authors should also provide data on Bergmann glial cells in the* Cxcr4-*CKO mutants. The authors state that* Cxcr4 *is absent from developing Bergmann glia (where is the data?). Is it possible that* FoxC1 *regulates Bergmann glial cells through a* Cxcr4-*independent pathway?*

We have added BLBP data in the *FoxC1* null mutant (Figure 2) and *Cxcr4* CKO (Figure 6), which we trust convey our conclusion more accurately. We show that Bergmann glia are present with abnormal morphology as seen in the paper published by the Littman group (Zou et al., Nature, 1998, Figure 4). The BG phenotype may be secondary to abnormal RG which express *Cxcr4*. While it may be possible that BG development is independent of *Cxcr4*, we see a similar phenotype in both *FoxC1*^*–/–*^ and *Cxcr4* CKOs. However, it is true that in the cerebellum the development of different cell types are dependent on one another. For example, Purkinje cells are closely associated with BG and PCs are required for BG function and survival. Hence disrupting one or more cell types could in turn affect other cell types, and therefore the effect could be indirect.

*5) The authors convincingly showed that cerebellar VZ development in* FoxC1a *and* Cxcr4 *mutants are grossly abnormal, with reduced number of proliferating cells and disorganized radial glial fibers. While this observation supports the model, it does not necessarily prove that the* FoxC1 *VZ phenotype is caused by demised Sdf1a expression in the meninges. One complication is that* FoxC1 *is required for development of meninges in the forebrain where it regulates production of RA for proper radial glial cell development (Cell, 139: 597, 2009). If development of cerebellar meninges is similarly disrupted, then the authors need to consider the possibility that VZ phenotype in* FoxC1 *mutants is caused by other factors in the meninges that are independent of Sdf1a.*

Unlike the *FoxC1* forebrain phenotype, meningeal development is normal in both *FoxC1* and *Cxcr4* mutants (Figure 1—figure supplement 1). We also observe the same phenotype in both *FoxC1*^*–/–*^ and *Cxcr4* CKOs. While other factors may contribute to both phenotypes, the major effector of *FoxC1* dependent signalling by the mesenchyme is SDF1α.

*6) In the* Cxcr4 *mutants, an interesting possibility is that Sdf1a secreted from other sources including the choroid plexus may activate* Cxcr4 *in the VZ. Therefore, the authors need to discuss this possibility and carefully examine the development of meninges and expression of Sdf1a in meninges of* FoxC1 *mutants. If meningeal development is compromised as in the forebrain, the authors need to perform* FoxC1 *rescue experiment (using exogenously supplied Sdf1a) to clearly support their model.*

In *Cxcr4* mutants, since the receptor gene has been knocked out of the brain, the presence of the ligand, SDF1α, will make no difference. We have previously reported a significant down regulation of SDF1α in the mesenchyme of *FoxC1* mutants. We did not conduct a rescue experiment since, unlike the forebrain, the meninges are structurally normal in *FoxC1* mutants at this stage, as evinced by Laminin staining (Figure 1—figure supplement 1).

[Editors’ note: the revised article was rejected after discussions between the reviewers, but a further resubmission was accepted after an appeal against the decision.]

*The reviewers feel that the revisions you have made have in part improved the manuscript. One reviewer was previously concerned about whether* FoxC1-*KO and* Cxcr4-*KO mutants have the same phenotype, including cell proliferation, Bergmann glia and Purkinje cell formation. The new data provided in the revision has partially addressed this concern. However all of the reviewers and the BRE feel that data relating to the meninges is lacking and severely reduces enthusiasm for the study. Although it is formally possible that* FoxC1-*KO disrupts other signaling molecules in the meninges, RA is an unlikely candidate. In the forebrain, loss of RA signaling due to* FoxC1-*KO blocks neurogenesis, which is very different from what is found in the cerebellum. As raised in the initial review, it is important to show normal formation of the meninges and the new data does not address this issue. This would require analysis with meninges markers. The reviewers are not convinced that the* FoxC1 *phenotype is caused by Sdf1 and whether Sdf1 expression is indeed regulated by* FoxC1*. There is no data confirming that Sdf1 expression is abolished in* FoxC1 *meninges. The cerebellum phenotype in* Cxcr4 *mutants seems to be similar to that of* FoxC1 *mutants. However, the* FoxC1 *mutant cerebellum is much more severe, with the basement membrane clearly detached from the pial surface (*Figure 1*), which would invariably disrupt Bergmann glial end feet attachment, likely causing their mis-localization. In summary, there is concern over major conclusions without a clear indication that meninges development is not compromised. In the absence of this evidence, the study cannot rule out the involvement of other factors derived from the meninges; hence the recommendation to reject the study.*

We request reconsideration of our work for the reasons described below. In addition, we have new data that easily address the remaining concerns.

Briefly, the reviewers concluded that our data did not fully support the major conclusion that *FoxC1* regulated cerebellar development via Sdf1a. This was based on 2 criticisms. We list these concerns and our response below:

1) *There is no data confirming that Sdf1 expression is abolished in* FoxC1^–/–^
*meninges.*

a) This is a factual error of review. We previously reported that Sdf1a is significantly downregulated at e12.5 in *FoxC1*^*–/–*^ posterior fossa mesenchyme compared to wild-type controls. This data is presented in Figure 4 of [3], *FOXC1* is required for normal cerebellar development and is a major contributor to chromosome 6p25.3 Dandy-Walker malformation, Nat Genet., 41: 1037–1042). We cited this reference and data multiple times throughout the manuscript. There may have been some confusion, since in the 2009 paper, Sdf1a was referred to as *Cxcl12*. Note that we did define all gene synonyms in our current manuscript, but this may have been overlooked.

b) Note also that we referenced [54], Meningeal defects alter the tangential migration of cortical interneurons in *FoxC1*hith/hith mice, Neural Dev., Jan 17; 7:2). In Figure 2 of the Zarbalis paper, the authors demonstrated that *FoxC1* directly regulates Sdf1a (*Cxcl12*) in the mouse meninges.

2) *“The reviewers are not convinced that the FoxC1 phenotype is caused by Sdf1a*.*”*

a) Our initial and revised submissions showed that loss of the Sdf1a receptor (*Cxcr4*) largely phenocopies the *FoxC1*^*–/–*^ cerebellar phenotype. Admittedly, the *FoxC1*^*–/–*^ phenotype is more severe, however, the phenocopy is striking and we addressed possible reasons in the Discussion.

To substantiate our model, we now have new, very strong rescue data as requested. Exogenously applied Sdf1a does indeed rescue the *FoxC1*^*–/–*^ phenotype.

b) The reviewers had concerns that the meninges was disrupted in *FoxC1*^*–/–*^. We concur that the mutant section in SFig1H did have a detached meninges. We missed this and apologize. However, this is not representative of *FoxC1* mutants, and we can easily replace that panel. We disagree that signal intensity varies and are in the midst of preforming additional staining for other markers.

Regardless, the fact that we see rescue of the earliest *FoxC1*^*–/–*^ cerebellar phenotype with Sdf1a, significantly reduces the relevance of these meningeal experiments.

In summary, we request that our *FoxC1*
^*–/–*^ work be considered for publication. Our findings are of interest to a broad basic and clinical scientific audience.